# Effects of Aviation Mutagenesis on Soil Chemical Indexes, Enzyme Activities, and Metabolites of Dahongpao (*Camellia sinensis*) Tea Trees

**DOI:** 10.3390/plants13101291

**Published:** 2024-05-08

**Authors:** Miao Jia, Pengyuan Cheng, Yuhua Wang, Xiaomin Pang, Mingzhe Li, Lei Hong, Qi Zhang, Yiling Chen, Xiaoli Jia, Jianghua Ye, Haibin Wang

**Affiliations:** 1College of Tea and Food, Wuyi University, Wuyishan 354300, China; 2College of Life Science, Longyan University, Longyan 364012, China; 3College of JunCao Science and Ecology, Fujian Agriculture and Forestry University, Fuzhou 350002, China

**Keywords:** aviation mutagenesis, *Camellia sinensis*, soil metabolites, soil enzymes, soil physicochemistry

## Abstract

Aviation mutagenesis is a breeding method for the rapid selection of superior plant varieties. In this study, rhizosphere soil chemical indexes, soil enzyme activities, and soil metabolites were measured in Dahongpao tea trees with aviation mutagenesis (TM) and without aviation mutagenesis (CK). The main soil metabolites distinguishing TM and CK and their relationships with soil chemical indexes and soil enzyme activities were analyzed and obtained. The results showed that there was no significant change in the rhizosphere soils’ pH of TM tea trees compared to CK (*p* = 0.91), while all other chemical indexes of TM were significantly higher than CK (*p* < 0.05). In addition, the activities of enzymes related to soil nutrient cycling such as urease, protease, sucrase, acid phosphatase and cellulase, and enzymes related to soil antioxidants such as superoxide dismutase, catalase, peroxidase, and polyphenol oxidase were significantly increased (*p* < 0.05) in the rhizosphere soils of TM tea trees compared to CK. Soil metabolite analysis showed that the main soil metabolites distinguishing CK from TM were carbohydrates, nitrogen compounds, and amines. Of these, carbohydrates and nitrogen compounds were significantly positively correlated with soil chemical indexes and soil enzymes, whereas amine was significantly negatively correlated with soil chemical indexes such as organic matter, total nitrogen, total potassium, available nitrogen, available phosphorus; amine showed significant negative correlation with soil enzymes such as catalase, peroxidase, polyphenol oxidase, and urease. It can be seen that aviation mutagenesis is conducive to improving the ability of tea tree rhizosphere aggregation and transformation of soil nutrients, increasing the total amount of soil nutrients and the content of available nutrients, which is more conducive to promoting the uptake of nutrients by the tea tree, and thus promoting the growth of the tea tree.

## 1. Introduction

Plant germplasm resources are an important basis for breeding new plant varieties and important materials for biological research. Aviation mutagenesis is a novel, rapid, and efficient breeding technology. Plant seeds in the process of aviation mutagenesis, through the integrated impacts of a powerful magnetic field, gravity, radiation, and other factors, are easily susceptible to change genetic and physiological characteristics, which in turn is conducive to the screening of high-quality germplasm resources [1,2]. For a long time, breeding mutagenesis of plant aviation has been greatly emphasized worldwide, and some plant varieties with special advantages have been obtained, e.g., varieties with high resistance, high yield, or excellent quality [3]. China attaches great importance to aviation mutagenesis breeding, and as of 2021, China has obtained more than 200 new plant varieties with excellent traits through aviation mutagenesis, which has made important contributions to the rapid development of Chinese agriculture [4]. Tea tree aviation mutagenesis breeding started late, in 2003, for the first time in China, when “Shenzhou V” took tea tree seeds into space for mutagenesis. Until 2023, China has sent 12 batches of different varieties of tea trees into space for mutagenesis breeding. However, there are still no reports on the effects of aviation mutagenesis on tea trees.

Wuyishan is an important tea-producing area in China and the birthplace of oolong tea. In November 2011, the seeds of Wuyi Rock Tea, which were ferried into space with the “Shenzhou VIII” spacecraft into space, underwent space mutation for a total of 16 days, 13 h, and 34 min, marking the unique event of being the only batch of Wuyi Rock Tea seeds to have been subjected to space mutation to date. In April 2012, the seeds of tea trees after aviation mutagenesis were germinated, nursed, and managed in the same plot as the control. A previous study found that the Dahongpao (*Camellia sinensis*) tea tree had already shown obvious differences in growth and physiology compared to the control after aviation mutagenesis, which was manifested by a significant increase in leaf length, leaf width, and leaf area, a significant increase in photosynthesis capacity, and a significant increase in quality index content [5]. It can be seen that aviation mutagenesis did have a significant effect on the growth and physiology of the Dahongpao tea tree.

Soil is the carrier of plant growth, the root is the organization of the plant in close contact with the soil, the nutrient conversion capacity of the rhizosphere soil determines the content of available nutrients in the soil, and affects the growth of the plant and its nutrient uptake [6]. Whereas soil nutrient conversion capacity was significantly and positively correlated with enzyme activities related to nutrient cycling in soil [7], Lin et al. [8] found that a decrease in the activity of nutrient transformation-related enzymes in the rhizosphere soil resulted in reduced available nutrient content in the soil, subsequently leading to a decline in tea yield and quality. The above changes are associated with changes in the composition and content of rhizosphere metabolites of tea trees [9]. During cultivation, plants affect the rhizosphere ecosystem by releasing secretions from the root system, which in turn alters the nutrient cycling of the rhizosphere soil to adapt to the environment [10,11]. In recent years, many scholars have analyzed the interactions between metabolites and environmental factors and their effects on soil function from the perspective of plant rhizosphere soil metabolites and have made important progress, concluding that the use of soil metabolomics technology can effectively analyze the functional changes of rhizosphere soils and their effects on soil nutrient cycling [12,13,14,15]. The authors of this study hypothesized that the growth and physiological advantages exhibited by Dahongpao tea trees after aviation mutagenesis may be closely related to changes in their rhizosphere soil microbial systems. Aviation mutagenesis may have altered the composition and rhizosphere metabolites content of tea trees, which in turn affected nutrient cycling capacity and available nutrient content in the soil, resulting in superior characteristics exhibited by the mutagenized Dahongpao tea trees. Accordingly, in this study, the rhizosphere soil chemical indexes, soil enzyme activities, and soil metabolites of Dahongpao tea trees with aviation mutagenesis (TM) and without aviation mutagenesis (CK) were measured to analyze and obtain the main soil metabolites distinguishing TM from CK and their relationship with soil chemical indexes and soil enzyme activities, so as to reveal the mechanism of the impacts of aviation mutagenesis on the growth of the Dahongpao tea tree, and to provide references for aviation breeding of tea trees and their dissemination.

## 2. Results

### 2.1. Analysis of Chemical Indexes of Tea Tree Rhizosphere Soil

An analysis of the rhizosphere soils’ chemical indexes of tea trees showed (Figure 1) that the rhizosphere soils’ pH of tea trees after aviation mutagenesis (TM) did not change significantly (*p* = 0.91) compared to tea trees without aviation mutagenesis (CK), whereas other chemical indexes were significantly higher than CK (*p* < 0.05). Specifically, the content of organic matter, total nitrogen, total phosphorus, total potassium, available nitrogen, available phosphorus, and available potassium in the rhizosphere soil of the tea tree increased from 7.72 to 11.28 g/kg, 0.45 to 1.11 g/kg, 0.30 to 0.80 g/kg, 4.02 to 5.29 g/kg, 4.72 to 7.01 mg/kg, 11.37 to 16.47 mg/kg, and 203.62 to 233.92 mg/kg, respectively, after aviation mutagenesis. It can be seen that aviation mutagenesis is conducive to increasing the total amount of rhizosphere soil nutrients and the content of available nutrient content, thus improving soil fertility.

### 2.2. Analysis of Rhizosphere Soil Enzyme Activities of Tea Tree

An analysis of the enzyme activities of tea tree rhizosphere soil showed (Figure 2A) that the activities of soil nutrient cycling-related enzymes such as urease, protease, sucrase, acid phosphatase, and cellulase in TM tea trees’ rhizosphere soils were significantly increased (*p* < 0.05) as compared to CK, from 1.67 to 2.03 U/g, 133.82 to 155.46 U/g, 113.37 to 140.36 U/g, 7.29 to 9.48 U/g, and 10.97 to 17.67 U/g, respectively. Further analysis found (Figure 2B) that the enzyme activities related to soil antioxidants, such as superoxide dismutase, catalase, peroxidase, and polyphenol oxidase, were also significantly increased (*p* < 0.05) in TM rhizosphere soils compared to CK from 11.25 to 15.27 U/g, 3.69 to 5.49 U/g, 4.88 to 5.89 U/g, and 2.53 to 5.65 U/g, respectively. It can be seen that aviation mutagenesis is conducive to improving enzyme activities related to nutrient cycling and antioxidants in the rhizosphere soils of tea trees, which in turn improves soil fertility and texture.

### 2.3. Analysis of Rhizosphere Soil Metabolites of Tea Tree

In this study, it was found (Figure 3A) that a total of 123 metabolites were detected in the rhizosphere soils of tea trees, of which there was no significant difference between TM and CK in the total amount of metabolites (*p* > 0.05). The principal component analysis showed (Figure 3B) that principal component 1 and principal component 2 could effectively distinguish TM and CK with an overall contribution of 73.4%. It can be seen that although there is no significant difference between TM and CK in the total amount of metabolites, there is some difference in the content of different metabolites. Accordingly, metabolites were further classified and analyzed by principal component analysis in this study, and the results showed (Figure 3C) that 123 metabolites could be classified into acids, carbohydrates, nitrogen compounds, lipids, amines, esters, alcohols, aromatics, phenols, ketones, hydrocarbons, heterocyclic compounds, aldehydes, and other categories. Of these, carbohydrates and nitrogen compounds were more relevant to TM, while esters, heterocyclic compounds, amines, aromatics, acids, and others were more relevant to CK.

In this study, we further constructed an orthogonal partial least squares discrimination analysis (OPLS-DA) model for CK and TM to screen for key metabolites with significant differences. The results of the post-construction effect evaluation of the OPLS-DA model showed (Figure 3D) that the fit of the model after 200 random simulations, R^2^Y = 0.998, reached a significant level (*p* < 0.005), and the predictability, Q^2^ = 0.879, also reached a significant level (*p* < 0.005). In addition, the score chart of the OPLS-DA model showed (Figure 3E) that CK and TM were clearly distinguished in different areas. It can be seen that the OPLS-DA model constructed in this study meets the requirements and can effectively distinguish CK from TM. Accordingly, this study further extracted 52 key metabolites from the OPLS-DA model that could effectively distinguish CK from TM (Figure 3F). The 52 key metabolites could be classified into 13 groups (Figure 3G), of which seven groups such as carbohydrates, nitrogen compounds, amines, alcohols, hydrocarbons, heterocyclic compounds, and ketones were significantly higher in their content in TM than CK; six groups such as esters, aromatics, lipids, acids, aldehydes, and others were significantly lower in their content in TM than CK. In this study, TOPSIS was used to analyze the influence weights of 13 groups of compounds in distinguishing CK and TM, and the results showed (Figure 3H) that four groups with influence weights greater than 10% were carbohydrates, nitrogen compounds, ester, and amine. It can be seen that the different types of metabolites in the rhizosphere soils of tea trees changed significantly in content after aviation mutagenesis.

### 2.4. Interaction Analysis

Based on the previous analysis, this study further analyzed the interactions of four groups of soil metabolites, which were significantly different in TM and CK, with chemical indexes and soil enzyme activities in the rhizosphere soils of tea trees. The results of a redundancy analysis between soil chemical indexes and four groups of soil metabolites showed (Figure 4A) that soil metabolites significantly associated with TM were carbohydrates, nitrogen compounds, and soil chemical indexes significantly associated were pH, organic matter, total nitrogen, total phosphorus, total potassium, available nitrogen, available phosphorus, and available potassium, while soil metabolites significantly associated with CK were ester and amine. The results of redundancy analysis of soil enzymes with four groups of soil metabolites showed (Figure 4B) that soil metabolites significantly associated with TM were carbohydrates and nitrogen compounds, and the soil enzymes significantly associated were urease, protease, sucrase, acid phosphatase, cellulase, superoxide dismutase, catalase, peroxidase, and polyphenol oxidase, while soil metabolites significantly associated with CK were ester and amine. The results of the correlation network analysis showed (Figure 4C) that both carbohydrates and nitrogen compounds were significantly and positively correlated with soil chemical indexes, except soil pH; both carbohydrates and nitrogen compounds were significantly and positively correlated with soil enzyme activities. In addition, amine was found to be significantly negatively correlated with organic matter, total nitrogen, total potassium, available nitrogen, and available phosphorus, while the correlation with other soil chemical indexes was not significant. Amine was significantly negatively correlated with catalase, peroxidase, polyphenol oxidase, and urease, while the correlation with other soil enzymes was not significant. There was no significant correlation between ester and either soil chemical indexes or soil enzymes. It can be seen that aviation mutagenesis significantly altered metabolites, enzyme activities, and soil nutrient contents of tea tree rhizosphere soil, especially with regards to increased accumulation of carbohydrates and nitrogen compounds, and decreased amine content; it also increased enzyme activity related to nutrient conversion and resistance in tea tree rhizosphere soil and increased the total amount of nutrients and the content of available nutrients in the soil.

## 3. Discussion

Soil is the carrier of plant growth and contains a large amount of nutrients needed for plant growth, and is an important place for plants to obtain nutrients [16]. The total amount of soil nutrients and their available nutrient content directly affect plant growth, especially the available nutrient content [17]. It has been reported that high soil nutrient transformation capacity is conducive to increasing the content of available nutrients, which in turn promotes plant nutrient uptake and plant growth [18]. In this study, it was found that organic matter, total nitrogen, total phosphorus, total potassium, available nitrogen, available phosphorus, and available potassium in the rhizosphere soils of tea trees after aviation mutagenesis (TM) were significantly higher than those of tea trees without aviation mutagenesis (CK). It can be seen that after aviation mutagenesis, the ability of nutrient aggregation and transformation by the rhizospheres of tea trees is enhanced, and the content of available nutrients is increased, which is more conducive to promoting the absorption of nutrients by tea trees, thus promoting the growth of tea trees.

Soil enzyme activity is an important index for assessing soil fertility and can reflect the effect of soil on plant growth [19]. Urease, protease, acid phosphatase, sucrase, and cellulase activities are closely related to soil nutrient cycling [7]. It has been reported that increasing soil urease and protease activities are conducive to improving soil nitrogen cycling and increasing soil available nitrogen content [20]. Increasing soil acid phosphatase activity is conducive to accelerating soil phosphorus transformation and increasing soil-available phosphorus content [21]. Increasing soil sucrase and cellulase activities are conducive to improving soil organic matter degradation and transformation and increasing soil nutrient content [22]. In this study, it was found that all enzyme activities related to nutrient cycling in TM tea trees’ rhizosphere soils were significantly higher than CK soil. In this study, it was found that the activity of antioxidant-related enzymes such as superoxide dismutase, catalase, peroxidase, and polyphenol oxidase was significantly enhanced in TM tea trees’ rhizosphere soils compared to CK. It has been reported that soil antioxidant enzyme activities are closely related to soil texture and play an important role in maintaining soil health [7]. Moreover, increasing the activity of soil antioxidant enzymes is conducive to promoting the multiplication of beneficial soil microorganisms, plant root growth, and nutrient uptake by plants [8]. It can be seen that changes in the rhizosphere soil environment of tea trees after aviation mutagenesis may be more conducive to the propagation of microorganisms in the soil, which will in turn increase the activity of antioxidant enzymes and nutrient cycling-related enzymes in the soil, improve soil texture, maintain soil health, promote soil nutrient cycling, and increase the soil nutrient content and its effectiveness.

Rhizosphere soil metabolomics can effectively analyze low molecular weight compounds in soil and is an important tool for assessing soil function [23]. In this study, we analyzed rhizosphere soil metabolites of TM and CK using metabolomics techniques and found that they differed significantly in the content of different types of metabolites. Further analysis using OPLS-DA and TOPSIS revealed that TM was significantly different from CK, and the key soil metabolites that effectively differentiated between the two were carbohydrates and nitrogen compounds, ester, and amine. Further interaction network analysis of these four groups of key metabolites with soil chemical indexes and soil enzymes revealed that carbohydrates and nitrogen compounds were significantly and positively correlated with soil chemical indexes, except soil pH, while amine was found to be significantly negatively correlated with organic matter, total nitrogen, total potassium, available nitrogen, and available phosphorus, and had insignificant correlation with other soil chemical indexes. There was no significant correlation between ester and either soil chemical indexes or soil enzymes. It is evident that the key metabolites, including carbohydrates, nitrogen compounds, and amine, are very important in differentiating and influencing TM and CK. It has been reported that carbohydrates serve as a microbial carbon source, and the increase in soil carbohydrate content facilitates microbial reproduction, accelerates soil organic matter degradation, and improves nutrient cycling capacity [24,25]. Nitrogen compounds are nitrogen sources, containing both organic and inorganic forms, and increasing their content is conducive to increasing the total amount of soil nutrients and laying the foundation for available nutrient transformation [26]. Amine has a significant effect on soil texture, and its accumulation in large quantities is highly likely to disrupt soil health, inhibit plant root growth, and hinder nutrient uptake by plant roots [27]. It can be seen that aviation mutagenesis increased the aggregation and accumulation of carbohydrates and nitrogen compounds in the rhizosphere soils of tea trees, while the opposite was true for amines. Aviation mutagenesis was conducive to promoting the propagation of soil microorganisms in the rhizosphere of tea trees, which was conducive to increasing the activity of enzymes related to nutrient cycling in the soil, thereby improving the accumulation and transformation of soil nutrients, increasing the total amount of nutrients and the content of available nutrients, and promoting the growth of tea trees.

In summary, this study analyzed the differences between unmutagenic and aviation mutagenic tea trees in their respective rhizosphere soils’ chemical indexes, soil enzyme activities, and soil metabolites. It was found that the total amount of soil nutrients and the content of available nutrients in the rhizosphere soil of tea trees after aviation mutagenesis were significantly increased, and the activities of nutrient cycling-related enzymes and antioxidant-related enzymes of their rhizosphere soil were significantly enhanced. In addition, it was found that the key metabolites distinguishing unmutagenic and aviation mutagenic tea trees in their rhizosphere soils were carbohydrates, nitrogen compounds, and amines, of which carbohydrates and nitrogen compounds were significantly positively correlated with soil chemical indexes and soil enzymes, while amine was significantly negatively correlated. It can be seen that aviation mutagenesis is conducive to improving the ability of aggregation and transformation of soil nutrients in the rhizosphere soil of tea trees, increasing the total amount of soil nutrients and the content of available nutrients, which is more conducive to promoting the uptake of nutrients by the tea tree, thus promoting the growth of tea trees. This study is of great significance for the development and popularization of aviation breeding of tea trees. However, how metabolites in tea trees’ rhizosphere soils affect soil microbial diversity and function after aviation mutagenesis, and then change nutrient transformation in the soil, requires further study.

## 4. Materials and Methods

### 4.1. Samples and Sampling

The sampling site of this study was located in “Xianmingyan Tea Factory Tea Tree Aerospace Breeding Experimental Base”, Wuyishan City, Nanping City, Fujian Province, China (117°59′47.7″ E 27°44′8.4″ N), and the sampled tea tree variety was Dahongpao (*Camellia sinensis*) with an age of 11 years. Tea trees were planted in acidic red loam soil. During the planting period, the tea trees were fertilized in October every year with compound fertilizer (N:P:K = 21:8:16) at a rate of 700 kg/ha. In May 2023, rhizosphere soils of Dahongpao tea trees with aviation mutagenesis (TM) and without aviation mutagenesis (CK) were collected to determine soil chemical indexes, soil enzyme activities, and soil metabolism. The rhizosphere soil sampling method of the tea tree [6] was expressed as randomly selecting five TM and CK tea trees, removing residues covering the soil surface, layer by layer shoveling the surface soil for about 35 cm, cutting the fine roots of the tea tree, gently shaking the root of the tea tree, and collecting the soil that was still adhered to the root, i.e., the rhizosphere soil of the tea tree. The rhizosphere soil of five tea trees was mixed to make an individual replicate, totaling about 150 g. Three independent replicates were taken for each sample.

### 4.2. Determination of Soil Chemical Indexes

The collected rhizosphere soils of the tea trees were used to determine soil chemical indexes, including pH, organic matter, total nitrogen, total phosphorus, total potassium, available nitrogen, available phosphorus, and available potassium. Three independent replicates were set up for each sample. For specific determination, refer to the method of Wang et al. [28]. Soil pH was determined using the potentiometric method, in which the ratio of water to soil was 2.5:1. Soil organic matter was determined by high-temperature oxidation of the soil using potassium dichromate and concentrated sulfuric acid, titration using ferrous sulfate solution, and conversion to organic matter content. Total nitrogen was determined by the Kjeldahl nitrogen method after high-temperature digestion of soil using concentrated sulfuric acid, and then converted to total nitrogen content. Total phosphorus and total potassium contents were determined by firstly dissolving the soil with NaOH, then using molybdenum antimony resistance colorimetry to determine total phosphorus content, and using the flame photometer method to determine total potassium content. Available nitrogen of the soil was determined by leaching the soil with a NaOH solution followed by hydrochloric acid titration of leachate and then converted to available nitrogen content. Available phosphorus was determined by a molybdenum antimony resistance colorimetric method after NaHCO_3_ extraction, and then converted to available phosphorus content. Available potassium was determined by a flame photometer method after extraction with ammonium acetate and then converted to available potassium content.

### 4.3. Determination of Soil Enzyme Activity

The collected tea tree rhizosphere soil was used to determine soil nutrient cycling-related enzymes and soil antioxidant-related enzyme activities using the Enzyme Linked Immunosorbent Assay Kit. Three independent replicates were performed for each sample. Soil nutrient cycling-related enzymes were mainly determined as urease, protease, sucrase, acid phosphatase, and cellulase; soil antioxidant-related enzymes were mainly determined as superoxide dismutase, catalase, peroxidase, and polyphenol oxidase. The assay method was briefly described as follows: 0.5 g of fresh soil was extracted using different soil enzyme Elisa enzyme immunoassay kits (Shanghai Preferred Biotechnology Co., Ltd., Shanghai, China), and the enzyme activity was determined using a multifunctional enzyme labeling instrument (BioTek Synergy2 Gene 5, Winooski, VT, USA), and the enzyme activity was expressed as U/g. The absorbance wavelengths of urease, protease, sucrase, acid phosphatase, cellulase, superoxide dismutase, catalase, peroxidase, and polyphenol oxidase are measured at 630 nm, 680 nm, 540 nm, 660 nm, 540 nm, 560 nm, 240 nm, 470 nm, and 430 nm, respectively.

### 4.4. Extraction and Determination of Soil Metabolites

An amount of 0.5 g of the collected rhizosphere soils of the tea trees were taken, and 1 mL of methanol:isopropanol:water (3:3:2, *V*/*V*/*V*) extract was added, vortexed, and oscillated for 3 min, and ultrasonicated for 20 min, and then centrifuged in a centrifuge pre-cooled to 4 °C at 12,000 r/min for 3 min, and the supernatant was collected. The supernatant was added to 0.02 mL of internal standard (10 μg/mL), blown dry under nitrogen, and then the sample was derivatized. The derivatization method was described as 0.1 mL of methoxyaminopyridine hydrochloride solution (0.015 g/mL), and was added to the samples and incubated at 7 °C for 2 h. Then 0.1 mL of BSTFA (containing 1% TMCS) was added and vortexed and shaken for 30 min. After derivatization, 0.2 mL of the liquid was taken and 1 mL of n-hexane was added, mixed, and passed through a 0.22 μm organic filter membrane for GC-MS determination.

The main equipment used for the GC-MS determination was an Agilent 8890 gas chromatograph coupled with a 5977B mass spectrometer and a DB-5MS column (30 m length × 0.25 mm i.d. × 0.25 μm film thickness, J&W Scientific, Santa Clara, CA, USA). The carrier gas was helium at a flow rate of 1.2 mL/min, and the sample volume was 1 μL with a 5:1 injection ratio in the pre-sample mode. The oven temperature was held at 40 °C for 1 min, and then raised to 100 °C at 20 °C/min, raised to 300 °C at 15 °C/min, and held at 300 °C for 5 min. All samples were analyzed in scan mode. The ion source temperature was 230 °C, and the ion transfer line temperature was 280 °C. Soil metabolites were qualified and quantified by selecting 2~3 qualitative ions and 1 quantitative ion for each compound, and the comparison database used was the NIST20 mass spectrometry database. After deducting the background, the selected ions all appeared in the mass spectrum and the retention time was in agreement with the standard reference value; then the compound was qualified, and then the compound was quantified by integrating and correcting according to the selected quantitative ions [29]. The key metabolites were screened using orthogonal partial least squares discrimination analysis (OPLS-DA), and those with variable importance for the projection (VIP) greater than 1 were identified to be key metabolites [30].

### 4.5. Statistical Analysis

Excel 2020 was used to perform preliminary statistics on raw data. Rstudio software (version 4.2.3) was used for graphic production of post-statistical data. The R packages used for box plots, principal component plots, OPLS-DA models, heat maps, redundancy analysis, and interaction network diagrams were gghalves 0.1.4, ggbiplot 0.55, ropls and mixOmics, pheatmap 1.0.12, vegan 2.6.4, and linkET 0.0.7.1, respectively. TOPSIS analysis was conducted in the SPSSAU online platform at https://spssau.com/ (accessed on 12 February 2024).

## Figures and Tables

**Figure 1 plants-13-01291-f001:**
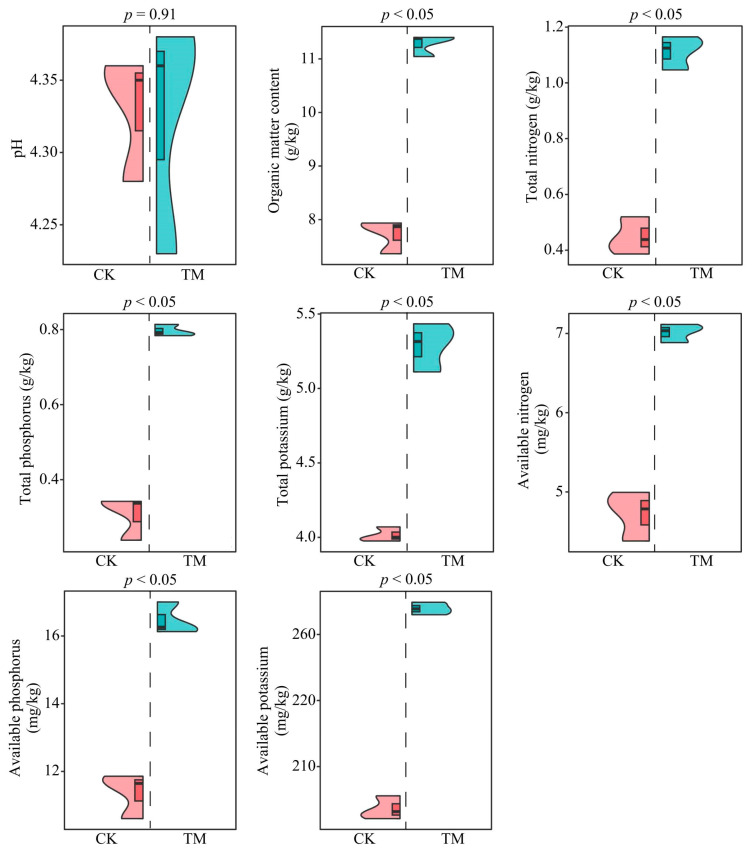
Effect of aviation mutagenesis on chemical indexes of the rhizosphere soils of tea trees. Note: CK: Unmutagenic Dahongpao tea tree; TM: Aviation mutagenic Dahongpao tea tree; the top of the violin-like graph is the maximum of the three repetitions, the bottom is the minimum, and the boxed horizontal line in the middle of the violin is the mean.

**Figure 2 plants-13-01291-f002:**
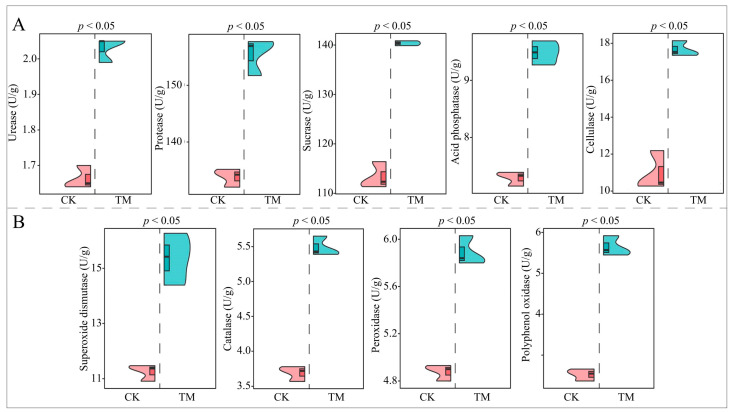
Effect of aviation mutagenesis on rhizosphere soils’ enzyme activities of tea trees. Note: CK: Unmutagenic Dahongpao tea tree; TM: Aviation mutagenic Dahongpao tea tree; (**A**): Changes of enzyme activity related to soil nutrient cycling; (**B**): Changes of antioxidant-related enzyme activities in soil; the top of the violin-like graph is the maximum of the three repetitions, the bottom is the minimum, and the boxed horizontal line in the middle of the violin is the mean.

**Figure 3 plants-13-01291-f003:**
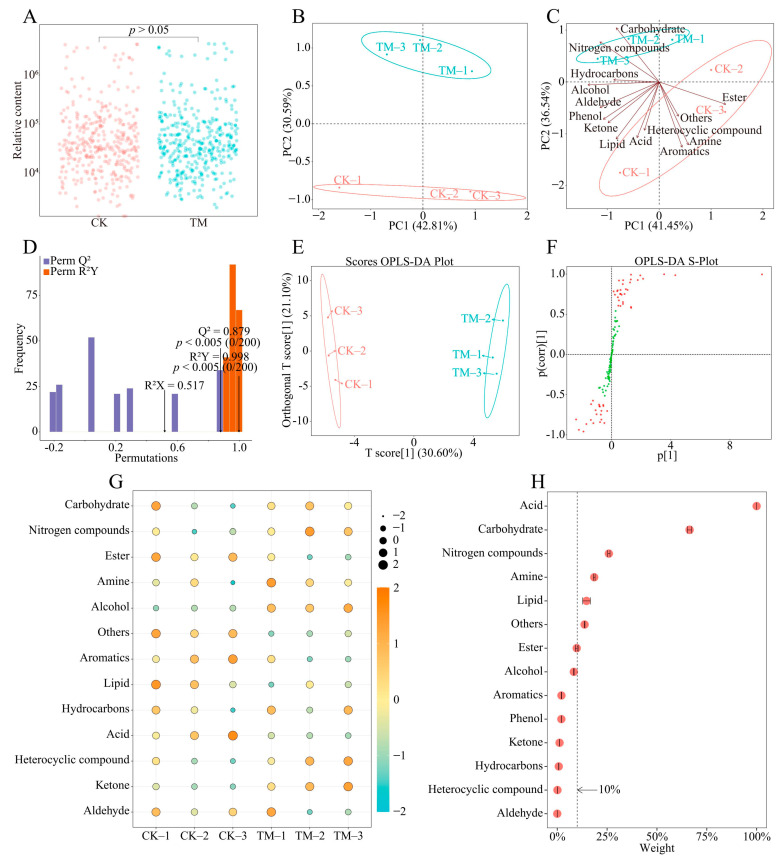
Effect of aviation mutagenesis on the rhizosphere soils’ metabolites of tea trees. Note: CK: Unmutagenic Dahongpao tea tree; TM: Aviation mutagenic Dahongpao tea tree; (**A**): content analysis of soil metabolites; (**B**): principal component analysis of soil metabolites; (**C**): principal component analysis of classified soil metabolites; (**D**): fit analysis of the OPLS-DA model constructed based on soil metabolites; (**E**): score plot analysis of the OPLS-DA model constructed based on soil metabolites; (**F**): S-Plot analysis of the OPLS-DA model constructed based on soil metabolites; (**G**): heat map analysis of different groups of key metabolites; (**H**): TOPSIS weighting analysis of different groups of key metabolites.

**Figure 4 plants-13-01291-f004:**
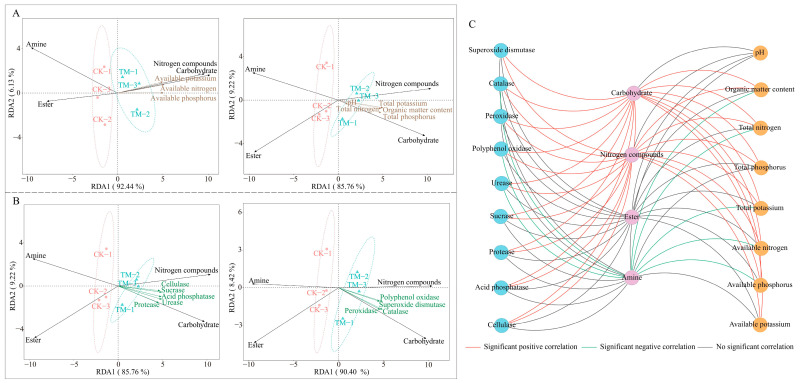
Interaction analysis of soil metabolites with soil chemical indexes and soil enzymes. Note: CK: Unmutagenic Dahongpao tea tree; TM: Aviation mutagenic Dahongpao tea tree; (**A**): redundancy analysis of soil metabolites with soil chemical indexes; (**B**): redundancy analysis of soil metabolites with soil enzymes; (**C**): correlation network analysis of soil metabolites with soil chemical indexes and soil enzymes.

## Data Availability

The data presented in this study are available as Appendix A.

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
