# Peer review of "Effects of Aviation Mutagenesis on Soil Chemical Indexes, Enzyme Activities, and Metabolites of Dahongpao (*Camellia sinensis*) Tea Trees"

_plants, 2024, doi:10.3390/plants13101291_

Round 1

Reviewer 1 Report

Comments and Suggestions for Authors

The manuscript titled “Effects of Aviation Mutagenesis on Physicochemical Indexes, Soil Enzyme Activities and Metabolites in the Rhizosphere Soil of Dahongpao (Camellia sinensis) Tea Trees” The study focused on rhizosphere soil physicochemical indexes, soil enzyme activities and soil metabolites were measured in Dahongpao tea trees with aviation mutagenesis (TM) and without aviation mutagenesis (CK). This study brings the mechanism behind the main soil metabolites distinguishing between TM and CK and their relationships with soil physicochemical indexes and soil enzyme activities. Why the author just has two treatments this is the main problem in the manuscript. The manuscript is not well written and organized, and it contains several repetitive contents or similar expressions. Therefore, it is hard for me to support publishing this study in the journal. Also, too many problems and the English language are needed so that the reviewers will easily understand your meaning. I suggest authors submit this manuscript to other relevant journals.

There are much-needed importance of clarity in the abstract and introduction with the fluency of research design and most importantly, the discussion portion needs major revision to link it with your findings and ground reality-based results of previous research.

The part of "Introduction" is not well organized, and the objective of the study is abrupt and not very clear to the readers and reviewers.

Language is just a medium that can convey your message spanning life and important moments of your life to find special findings, so I emphasize the authors to improve the English language and grammar of the article and try to make the explanation as simple as you can to attract new and non-familiar or non-native readers.

 Evaluate the results and link them all in a complimentary manner so that a clear picture draws your efforts for solving the problem in this research domain.

The novelty of the study needs to be highlighted compared to other similar studies.

Discussion is weak. The discussion needs enhancement with real explanations not only agreements and disagreements because of analysis nothing more. Authors should improve it by the demonstration of biochemical/physiological causes of obtained results. Instead of just justifying results, results should be interpreted, and explained to appropriately elaborate inferences.

Discussion seems to be poor, did not give good explanations of the results obtained. I think that it must be improved. Where possible please discuss potential mechanisms behind your observations. You should also expand the links with prior publications in the area, but try to be careful to not over-reach. For the latter, you should highlight potential areas of future study. I strongly recommend the rejection.

Comments on the Quality of English Language

Language is just a medium that can convey your message spanning life and important moments of your life to find special findings, so I emphasize to the authors to improve the English language and grammar of the article and try to make the explanation as simple as you can to attract new and non-familiar or non-native readers.

Author Response

Comments and Suggestions for Authors

The manuscript titled “Effects of Aviation Mutagenesis on Physicochemical Indexes, Soil Enzyme Activities and Metabolites in the Rhizosphere Soil of Dahongpao (Camellia sinensis) Tea Trees” The study focused on rhizosphere soil physicochemical indexes, soil enzyme activities and soil metabolites were measured in Dahongpao tea trees with aviation mutagenesis (TM) and without aviation mutagenesis (CK). This study brings the mechanism behind the main soil metabolites distinguishing between TM and CK and their relationships with soil physicochemical indexes and soil enzyme activities. Why the author just has two treatments this is the main problem in the manuscript.

The manuscript is not well written and organized, and it contains several repetitive contents or similar expressions. Therefore, it is hard for me to support publishing this study in the journal. Also, too many problems and the English language are needed so that the reviewers will easily understand your meaning. I suggest authors submit this manuscript to other relevant journals.

There are much-needed importance of clarity in the abstract and introduction with the fluency of research design and most importantly, the discussion portion needs major revision to link it with your findings and ground reality-based results of previous research.

A: Thank you very much to the reviewers. Regarding only two treatments in the manuscript, firstly, it is very rare for tea trees to be sent to space for aviation mutagenesis, and furthermore, this study used tea trees planted from the same batch of seeds, some of which underwent aviation mutagenesis and others did not, so it was reasonable to select Dahongpao tea trees with aviation mutagenesis (TM) and without aviation mutagenesis (CK) for the study. Thanks to the reviewers.

Secondly, the authors also carefully revised the manuscript and adjusted its structure and organization. Regarding the English language aspect, the authors have asked Lily, a colleague from a native English-speaking country, who has been engaged in related research field for more than ten years, to make a comprehensive revision of the English language. We hope that the manuscript will meet the requirements of the reviewers.

The part of "Introduction" is not well organized, and the objective of the study is abrupt and not very clear to the readers and reviewers.

Language is just a medium that can convey your message spanning life and important moments of your life to find special findings, so I emphasize the authors to improve the English language and grammar of the article and try to make the explanation as simple as you can to attract new and non-familiar or non-native readers.

A: Thanks to the reviewing experts. Based on the reviewer's comments, the authors have combed the introduction section. The main contents include the following: 1. Explain the importance of aviation mutagenesis in plant breeding and the results it has achieved. 2. Explain the application of aviation mutagenesis in the breeding of tea tree. 3. point out that there is an extreme lack of research on tea trees after aviation mutagenesis, which has not been reported so far. 4. point out that the object of this study is the only batch of tea trees of Wuyi rock tea that have undergone aviation mutagenesis. 5. Explain that it has been found in the previous study that the   tea trees with aviation mutagenesis have undergone significant changes and showed obvious advantages in growth. 6. Point out that the reasons for tea trees with aviation mutagenesis to have such advantages may be related to their nutrient uptake and metabolite changes in the soil. 7. Citing relevant literature, it was shown that there is a correlation between changes in soil metabolites and changes in soil nutrients, which may lead to changes in the growth of tea trees. 8. It was hypothesized that after aviation mutagenesis, there may be a relationship between the growth dominance of tea trees and the effectiveness of their characteristic metabolites and nutrients in the soil. 9. Accordingly, the study was carried out.

 Evaluate the results and link them all in a complimentary manner so that a clear picture draws your efforts for solving the problem in this research domain.

The novelty of the study needs to be highlighted compared to other similar studies.

A: Thanks to the reviewers for their suggestions. The authors have combed the results section to describe the differences between tea trees with aviation mutagenesis and without aviation mutagenesis from data obtained from experimental research. Describe the results from the facts, mainly including: 1. Describe the differences between CK and TM in rhizosphere soil physicochemical indexes; 2. Describe the differences between CK and TM in rhizosphere soil enzyme activities; 3. Describe the differences between CK and TM in rhizosphere soil metabolites, and screen for the key differential metabolites; and 4. Comprehensively analyze the relationship between the key metabolites, soil physicochemical indexes, and soil enzyme activities.

Discussion is weak. The discussion needs enhancement with real explanations not only agreements and disagreements because of analysis nothing more. Authors should improve it by the demonstration of biochemical/physiological causes of obtained results. Instead of just justifying results, results should be interpreted, and explained to appropriately elaborate inferences.

 Discussion seems to be poor, did not give good explanations of the results obtained. I think that it must be improved. Where possible please discuss potential mechanisms behind your observations. You should also expand the links with prior publications in the area, but try to be careful to not over-reach. For the latter, you should highlight potential areas of future study. I strongly recommend the rejection.

A: Thank you to the reviewers. The authors have revised the discussion section as appropriate. The authors briefly describe the results obtained, while citing relevant literature for elaboration, and suggest reasons for the changes in different indexes of rhizosphere soils of tea trees after aviation mutagenesis. Secondly, through the results obtained from the study, the authors also proposed the direction and content that should be studied in depth in the subsequent research, with a view to better revealing the reasons for the superiority of the growth of tea trees after aviation mutagenesis.

Comments on the Quality of English Language

Language is just a medium that can convey your message spanning life and important moments of your life to find special findings, so I emphasize to the authors to improve the English language and grammar of the article and try to make the explanation as simple as you can to attract new and non-familiar or non-native readers.

A: Thank you to the reviewers. The authors have asked Lily, a colleague from a native English-speaking country, who has been engaged in related research field for more than ten years, to make a comprehensive revision of the English language. We hope that the manuscript will meet the requirements of the reviewers.

Reviewer 2 Report

Comments and Suggestions for Authors

Review_ Plants-2940460

Thank you for the opportunity to review this paper.

The identification of new analysis methods for obtaining food products with superior properties is a permanent concern. The tea tree is such a product due to its use in China and the world.

In the Materials and Methods chapter, the study methods proposed by the authors are presented, methods that were chosen correctly and in accordance with the purpose of the work. The aim of the work was clearly stated, and the identification of analysis methods to achieve the proposed purpose were carefully selected. In this sense, the physico-chemical and enzymatic parameters at the soil level, respectively the metabolites, were correctly identified.

The obtained results are presented in detail; the statistical analysis of the obtained data is well documented. The presentation of the research results was made through numerous figures that clearly explain the distribution of fizico-chemical parameter, enzimatic activities and metabolites counts, respectively their corelation and their implications. Based on the analysis of the parameters, it was highlighted that aviation mutagenesis has favorable effects on the ability to aggregate and transform nutrients from the soil, increasing the total amount of nutrients and nutrients available from the soil, determining favorable effects for the development of the tea tree.

The Discussion chapter reports its own results to existing data in the literature. The manuscript is large and complex, it is well scientifically documented.

The conclusions established by the authors are clear and complex, all the results obtained as a result of the study encourage the use of aviation mutagenesis in promoting the growth of the tea tree.

I recommend that this paper be accepted and published in this journal.

Author Response

Comments and Suggestions for Authors

Review_ Plants-2940460

Thank you for the opportunity to review this paper.

The identification of new analysis methods for obtaining food products with superior properties is a permanent concern. The tea tree is such a product due to its use in China and the world.

In the Materials and Methods chapter, the study methods proposed by the authors are presented, methods that were chosen correctly and in accordance with the purpose of the work. The aim of the work was clearly stated, and the identification of analysis methods to achieve the proposed purpose were carefully selected. In this sense, the physico-chemical and enzymatic parameters at the soil level, respectively the metabolites, were correctly identified.

The obtained results are presented in detail; the statistical analysis of the obtained data is well documented. The presentation of the research results was made through numerous figures that clearly explain the distribution of fizico-chemical parameter, enzimatic activities and metabolites counts, respectively their corelation and their implications. Based on the analysis of the parameters, it was highlighted that aviation mutagenesis has favorable effects on the ability to aggregate and transform nutrients from the soil, increasing the total amount of nutrients and nutrients available from the soil, determining favorable effects for the development of the tea tree.

The Discussion chapter reports its own results to existing data in the literature. The manuscript is large and complex, it is well scientifically documented.

The conclusions established by the authors are clear and complex, all the results obtained as a result of the study encourage the use of aviation mutagenesis in promoting the growth of the tea tree.

I recommend that this paper be accepted and published in this journal.

A: Many thanks to the reviewers. Thank you for recognizing the work of our research team. Our research team has a reasonable staff structure and close cooperation. In recent years, we have published more than fifty papers in the field of tea science and technology, of which more than 20 have been published in Q1 and above journals. The team members have successively won the title promotion, scientific research honors, project projects and so on. In brief, our research team is a dynamic and excellent team. Thanks again for the recognition.

Reviewer 3 Report

Comments and Suggestions for Authors

This research is highly novel because it examines carbon compounds in the rhizosphere soil of tea plants bred by a special breeding method called Aviation mutagenesis. The science field of this research is suitable for publishing in this journal. The meta-analysis of soil nutrients, soil enzyme activity, and soil extracts is considered to be sufficient. However, I believe that further modifications are necessary for this paper to be accepted. Please consider the following points

The term "physicochemical indexes" is used too often, but the measurement items are simple and are merely "chemical indexes", so please correct this.

Please provide simple data on the good growth characteristics of the TM tea trees themselves.

The texts in most of the figures are too small and difficult to see. Please make them reasonably large.

TThe CK and TM samples have only 3 iterations each. Figure 3g shows that there is a large variation in the concentration of organic compounds in the soil. I’m wondering that there are the significant differences in most of the data as shown in Figures 1 and 2.

The x-axis are not described in each of the graphs in Figures 1 and 2. Please add it. Also, please add a footnote explaining the box-and-whisker diagram, which is shown in the graphs.

There is no explanation on soil where tea trees are grown. Please add information on how the soil is managed, including fertilizer application and soil properties.

The Discussion section is redundant, as it contains many of the same information described in the Results section. Please provide more interpretation of the data by discussing in depth the relevance of the results of this study and comparing them to previous studies. Rhizosphere soils contain substances secreted by plant roots and metabolites synthesized by soil microorganisms. Please include a discussion of this as well. In particular, please discuss soil enzyme production by soil microorganisms, as many of the soil enzymes are thought to be derived from soil microorganisms.

Author Response

Comments and Suggestions for Authors

This research is highly novel because it examines carbon compounds in the rhizosphere soil of tea plants bred by a special breeding method called Aviation mutagenesis. The science field of this research is suitable for publishing in this journal. The meta-analysis of soil nutrients, soil enzyme activity, and soil extracts is considered to be sufficient. However, I believe that further modifications are necessary for this paper to be accepted. Please consider the following points

The term "physicochemical indexes" is used too often, but the measurement items are simple and are merely "chemical indexes", so please correct this.

A: Thank you to the reviewers. The authors have revised it.

Please provide simple data on the good growth characteristics of the TM tea trees themselves.

A: Thanks to the reviewers. The indexes of TM growth superiority and the data at the seedling stage have been published. The authors have already mentioned it in the introduction. The growth indexes of tea tree corresponding to the period in this study were used by the authors for other manuscript entitled “Changes in growth and physiological property of tea tree after aviation mutagenesis and screening and functional verification of its characteristic hormones”. The above manuscript has been submitted to the journal “Horticultural Plant Journal”. Re-introducing the results into this manuscript would result in data reuse.

The authors list the results described in the manuscript that have been submitted as follows. “The results showed that leaf length, leaf width and leaf area of the 1st to 3rd leaves of TM tea trees were significantly greater than those of CK. Moreover, the surface of the leaves of TM tea tree is smoother and the edges are more closely and regularly serrated. It can be seen that the morphology of tea tree leaves changed significantly after aviation mutagenesis. Further analysis of growth indexes and yield of tea trees revealed that leaves number, bud density, hundred-bud weight, leaf area, and yield of TM tea trees were significantly higher than those of CK from 3.25 to 5.25 leaves, 3.50 to 5.25 103/m2, 70.20 to 75.66 g, 30.09 to 38.16 cm2 and 4412.50 to 6482.25 kg/hm2 , respectively. It can be seen that after aviation mutagenesis, the leaf morphology of the tea tree changed significantly, the growth indexes rose significantly, the yield increased and showed more superior traits. ”.

The texts in most of the figures are too small and difficult to see. Please make them reasonably large.

A: Thank you to the reviewer. The authors have made revisions to the font of the figure in the manuscript.

The CK and TM samples have only 3 iterations each. Figure 3g shows that there is a large variation in the concentration of organic compounds in the soil. I’m wondering that there are the significant differences in most of the data as shown in Figures 1 and 2.

A: Thank you to the reviewer. The difference between the data in Figures 1 and 2 is shown in the figures. Figures 1 and 2 are violin diagrams, with the maximum of 3 repetitions at the top of the violin, the minimum at the bottom, and the boxed horizontal line in the middle of the violin showing the mean. Thus, the overall repeatability is better as can be seen from the figure.

The x-axis are not described in each of the graphs in Figures 1 and 2. Please add it. Also, please add a footnote explaining the box-and-whisker diagram, which is shown in the graphs.

A: Thank you to the reviewer. The X-axis in Figures 1 and 2, the authors are putting it in the graph. At the request of the expert, the author takes it out and puts it outside. Secondly, the author added an explanation of the violin in the graph.

There is no explanation on soil where tea trees are grown. Please add information on how the soil is managed, including fertilizer application and soil properties.

A: Thank you to the reviewers. The authors added the nature of the soil and the timing, amount and type of fertilizer in the materials and methods.

The Discussion section is redundant, as it contains many of the same information described in the Results section. Please provide more interpretation of the data by discussing in depth the relevance of the results of this study and comparing them to previous studies. Rhizosphere soils contain substances secreted by plant roots and metabolites synthesized by soil microorganisms. Please include a discussion of this as well. In particular, please discuss soil enzyme production by soil microorganisms, as many of the soil enzymes are thought to be derived from soil microorganisms.

A: Thanks to the reviewers. The authors have revised the discussion appropriately. The results section has been simplified and analyzed in depth by the authors for discussion purposes. Hopefully, it will meet the requirements.

Round 2

Reviewer 1 Report

Comments and Suggestions for Authors

The author has improved the article and I recommend accepting it in the current form. 

Author Response

Comments and Suggestions for Authors

The author has improved the article and I recommend accepting it in the current form. 

A: I am grateful to the reviewing experts for their suggestions, which led to an effective improvement in the quality of the manuscript. Thanks again to the reviewing experts.

Reviewer 3 Report

Comments and Suggestions for Authors
  • This manuscript has been properly revised according to the comments.
  • I recommend publication with only a modest amount of revision as per my comments.
  • The soil enzymes Phosphatases (acid and alkaline phosphatases), Phenol oxidases, Ureases, Dehydrogenases, Proteases, and Cellulases have been reported to be produced by soil microorganisms . As mentioned in my previous review, it is likely that the TM tea trees did not produce these soil enzymes, but rather soil microorganisms did. the TM tea trees are described as having produced these soil enzymes, but there are no data to indicate this. This could be misleading to the reader, who should assume that TM tea trees create a rhizosphere environment that allows soil microorganisms that produce these soil enzymes to produce.

Author Response

Comments and Suggestions for Authors

This manuscript has been properly revised according to the comments.

I recommend publication with only a modest amount of revision as per my comments.

The soil enzymes Phosphatases (acid and alkaline phosphatases), Phenol oxidases, Ureases, Dehydrogenases, Proteases, and Cellulases have been reported to be produced by soil microorganisms . As mentioned in my previous review, it is likely that the TM tea trees did not produce these soil enzymes, but rather soil microorganisms did. the TM tea trees are described as having produced these soil enzymes, but there are no data to indicate this. This could be misleading to the reader, who should assume that TM tea trees create a rhizosphere environment that allows soil microorganisms that produce these soil enzymes to produce.

A: Many thanks to the reviewers. The authors have added to the discussion that aviation mutagenesis may be more beneficial in promoting microbial propagation in the soil, which in turn improves soil enzyme activity. Thanks again to the reviewing experts.